# Functional Recovery Caused by Human Adipose Tissue Mesenchymal Stem Cell-Derived Extracellular Vesicles Administered 24 h after Stroke in Rats

**DOI:** 10.3390/ijms222312860

**Published:** 2021-11-28

**Authors:** Francieli Rohden, Luciele Varaschini Teixeira, Luis Pedro Bernardi, Pamela Cristina Lukasewicz Ferreira, Mariana Colombo, Geciele Rodrigues Teixeira, Fernanda dos Santos de Oliveira, Elizabeth Obino Cirne Lima, Fátima Costa Rodrigues Guma, Diogo Onofre Souza

**Affiliations:** 1Graduate Program in Biological Sciences: Biochemistry, Department of Biochemistry, Universidade Federal do Rio Grande do Sul—UFRGS, Porto Alegre 90040-60, Brazil; luciele.sm@gmail.com (L.V.T.); luispedrobernardi2@gmail.com (L.P.B.); pamlukasewicz@gmail.com (P.C.L.F.); fatima.guma@ufrgs.br (F.C.R.G.); 2Instituto de Cardiologia do Rio Grande do Sul Fundação Universitária de Cardiologia, Porto Alegre 90620-101, Brazil; 3Faculty of Biomedicine, Universidade Federal de Ciências da Saúde de Porto Alegre—UFCSPA, Porto Alegre 90050-170, Brazil; 4Faculty of Pharmacy, Universidade Federal do Rio Grande do Sul—UFRGS, Porto Alegre 90040-60, Brazil; colombo_mari@hotmail.com; 5Experimental Research Center, Reproductive and Cellular Pharmacology Laboratory, Hospital de Clínicas de Porto Alegre, Porto Alegre 90035-903, Brazil; grteixeira@hcpa.edu.br (G.R.T.); fesoliveira@hcpa.edu.br (F.d.S.d.O.); cirnelima@hcpa.edu.br (E.O.C.L.)

**Keywords:** human adipose tissue mesenchymal stem cells, extracellular vesicles, experimental ischemic stroke, intranasal treatment, neuroprotection, functional recovery

## Abstract

Ischemic stroke is a major cause of death and disability, intensely demanding innovative and accessible therapeutic strategies. Approaches presenting a prolonged period for therapeutic intervention and new treatment administration routes are promising tools for stroke treatment. Here, we evaluated the potential neuroprotective properties of nasally administered human adipose tissue mesenchymal stem cell (hAT-MSC)-derived extracellular vesicles (EVs) obtained from healthy individuals who underwent liposuction. After a single intranasal EV (200 µg/kg) administered 24 h after a focal permanent ischemic stroke in rats, a higher number of EVs, improvement of the blood–brain barrier, and re-stabilization of vascularization were observed in the recoverable peri-infarct zone, as well as a significant decrease in infarct volume. In addition, EV treatment recovered long-term motor (front paws symmetry) and behavioral impairment (short- and long-term memory and anxiety-like behavior) induced by ischemic stroke. In line with these findings, our work highlights hAT-MSC-derived EVs as a promising therapeutic strategy for stroke.

## 1. Introduction

Stroke is a major cause of morbidity and mortality worldwide and affects health, economic, and social capabilities [1,2,3]. Ischemic stroke represents ~85% of total strokes [2], frequently causing neurological and neuropsychiatric sequelae [3,4,5,6]. In addition, clinical observations have shown that 20–50% of patients experience memory disorders [7].

The pathophysiology of ischemic stroke is characterized by blood flow obstruction in a restricted brain region, forming an infarct nucleus surrounded by an area known as the penumbra zone (known as the peri-infarct region in animal models) [8], which is an affected zone that can be recovered [9]. Reperfusion of the penumbra/peri-infarct zone contributes to a reduction in the final infarct size and attenuation/reversal of neurological and behavioral deficits [8]. The gold standard treatment for ischemic stroke involves thrombolytic strategies, which focus on optimizing the reperfusion time in the penumbra zone [10,11]. These strategies require qualified professionals and infrastructures capable of performing imaging studies such as sophisticated and expensive neuroimaging exams [12,13,14]. Therefore, innovative pharmacological therapeutic strategies focusing on salvaging the penumbra zone have been intensively investigated [11,15,16,17,18,19,20,21,22]. Accordingly, this study aimed to develop a new and accessible pharmacological strategy to promote functional recovery after focal permanent ischemic stroke in rats.

In vivo studies using rat models of brain ischemia demonstrated that systemic treatment with mesenchymal stem cells (MSC) improves neurological outcome and alleviates the symptoms of disability in therapeutic intervention performed 12 and 24 h after an ischemic insult [23,24]. Although MSC therapy seems promising, improvements have been made to optimize their effectiveness [25]. It is currently thought that the protective effects of MSCs are due to the release of extracellular vesicles (EVs) [26,27,28]. EVs are small double-membrane vesicles (30–200 nm) released by many cell types involved in mediating cell-to-cell communication under physiological conditions [29,30,31,32]. Compared to MSC, it has been shown that EV treatment is more effective in decreasing the risk of obstructive vascular effects and secondary microvascular thrombosis, while presenting a higher ability to cross the blood–brain barrier (BBB) [31,33,34,35]. Regarding immunological rejection, EVs exhibiting major histocompatibility complex class II (MHC-II) and other costimulatory molecules do not stimulate T cells [32,36], and there is evidence that MSC-derived EVs have lower immunogenic responses than MSCs [37]. In addition, Dabrowska et al., 2019 [26], demonstrated that EVs derived from human bone marrow mesenchymal stem cells used in the treatment of stroke in rats were able to decrease the levels of pro-inflammatory cytokines involved in the graft rejection process. In previous experimental studies, EVs systemically administered after ischemic stroke increased angiogenesis and neurogenesis and attenuated neuroinflammation evoked by focal brain injury [34,38]. However, there are reports that systemically administered EVs are located in organs other than the brain [39,40]. Thus, some studies have focused on a straightforward and noninvasive strategy to administer EVs specifically targeting the brain, such as intranasal administration, as a potential strategy to treat brain disorders [41,42,43].

A set of methodologies was performed, aiming to present original contributions and results to the understanding of certain components of stroke and of neuroprotective strategies of EVs. Specifically, we evaluated the short- and long-term effects on forepaws symmetry, behavioral performance, and anxiety-like behavior; short effect on EVs distribution in parenchyma brain, brain infarct volume and BBB permeability; and long-term effect on brain angiogenesis. MSC were obtained from human adipose tissue mesenchymal cells (hAT-MSC) through a minimally invasive procedure (liposuction) from healthy individuals. Additionally, intranasal EVs administration presented neuroprotective effect even when administered 24 h after stroke (200 µg/kg), this effect was effective for up to 42 days after stroke. This neuroprotective effect may be related to the high number of EVs in the peri-infarct region when compared to other brain regions. 

## 2. Results

### 2.1. Characterization of hAT-MSCs and EVs 

#### 2.1.1. hAT-MSC Characterization

Figure 1 shows the representative images of cell characterization. There is no single marker for hAT-MSCs; thus, they are characterized by immunophenotyping based on the presence (>70%) of cluster of differentiation (CD)90 and CD105 associated with the absence (<5%) of CD34 and CD45 [44]. Three sources of hAT-MSCs were used in this study, namely Cell (C) 0 (commercial, Figure 1A), C1 (donor 1, Figure 1B), and C2 (donor 2, Figure 1C), which were characterized by flow cytometry and fluorescence microscopy. In flow cytometry, the displacement of the fluorescence peak on the right side showed a positive value for the presence of protein markers. More than 70% of the C0, C1, and C2 were positive for CD90 and CD105 (Figure 1(Aa,Ba,Ca)), while only 0.3% of the analyzed cells were positive for CD45 and CD34 (Figure 1(Ab,Bb,Cb)), which is a characteristic of hAT-MSCs. The cells were also characterized by fluorescence microscopy using the same markers (Figure 1(A(c,d),B(c,d),C(c,d))).

#### 2.1.2. EVs Characterization

Figure 2 shows the three sources of EVs used in this study, namely, EV0 obtained from C0 (Figure 2A), EV1 obtained from C1 (Figure 2B), and EV2 obtained from C2 (Figure 2C), which were characterized using four protocols. EVs were detected inside the cells by confocal microscopy through the presence of CD63 and CD81 EV markers (Figure 2(Aa,Ba,Ca)). Released EVs were analyzed using flow cytometry; more than 90% of the EVs presented CD63 and CD81 [45,46]. Histograms are presented for EV0 (Figure 2(Ab)), EV1 (Figure 2(Bb)), and EV2 (Figure 2(Cb)). The diameter of the released EVs was confirmed using the Transmission electronic microscopy (TEM)-direct technique (EV0 Figure 2(Ac), EV1 Figure 2(Bc), and EV2 Figure 2(Cc)); the EVs suspension consisted only of vesicles with a cylindrical morphology and electron-dense membranes. The Zetasizer instrument was used to measure an average diameter of 140 nm, with a PDI average of 0.3 for EV0 (red), EV1 (green), and EV2 (blue) (Figure 2D).

### 2.2. Front Paws Symmetry 

#### Effect of Stroke on Paws Symmetry

Figure 3 shows that all animals were subjected to the cylinder test (CT) 24 h before stroke (day-1), and only animals with ~100% front paws symmetry before surgery were included in the study. Seventy-two hours after stroke, all ischemic (ISC) groups (treated and untreated) presented a mean symmetry of ~30%.

Dose curve effect of EV0 treatment on front paws symmetry. Figure 3a shows the dose curve used to determine the lowest dose for EV treatment. Intranasal administration of EV0 or vehicle was performed 24 h after stroke with the naive group as control. Animals treated with EV0 (100, 200, or 300 µg/kg) showed a time- and dose-dependent better improvement in front paws symmetry (from day seven after treatment) compared to the ischemic untreated animals; however, only animals treated with 200 or 300 µg/kg EVs attained symmetry scores similar to those of the naive group (total recovery). Thus, a dose of 200 μg/kg was used in further experiments. 

Effect of 200μg/kg EV treatment on front paws symmetry: Figure 3b shows that EV0, EV1, and EV2 treatments applied 24 h after a stroke caused better improved time-dependent recovery of front paws symmetry compared to the ISC group (from day seven after treatment). Total recovery was attained on day 28 for all EVs. Appendix A shows individual animal symmetry values of each EV treatment effect compared to the ISC group. 

### 2.3. Infarct Volume and BBB Permeability

Figure 4 shows that the infarct volume (2,3,5-Triphenyltetrazolium chloride-TTC staining) and BBB permeability (Evans blue penetration into the brain and spinal cerebrospinal fluid (CSF) albumin levels) were evaluated 72 h after the ischemic insult (48 h after EV0 or EV2 treatment). Treatment with 200 μg/kg EVs significantly decreased infarct volume and partially recovered Blood–Brain Barrier (BBB) permeability. Figure 4a (slice 0 mm to Bregma) and Figure 4b indicate the infarct volume. Figure 4c–e. indicate the Evans blue penetration into the brain parenchyma (Figure 4c (whole brain) and Figure 4d (slice 0 mm to Bregma); Figure 4e quantification of Evans Blue-EB in the brain). Figure 4f shows the CSF albumin levels.

### 2.4. EVs Distribution in the Brain Parenchyma

Figure 5 shows the distribution of EV0 (200 µg/kg) in the cerebral cortex evaluated by images acquired in three positions (Figure 5a). The regions examined were the supplementary motor cortex (M2) and somatosensory (SS) regions, both in the ipsi- and contra-lateral hemispheres. Remarkably, there was no homogeneous distribution of EVs throughout the brain. In the ISC group, the M2-ipsilateral (M2-I) and M2-contralateral (M2-C) regions presented a greater number of vesicles compared to the naive animals and to the ISC SS-ipsilateral (SS-I) and ISC SS-contralateral (SS-C) regions (Figure 5b). Figure 5c–h shows representative images of these findings.

### 2.5. Open Field Task (OFT)

Figure 6 shows that the same animals were subjected to all three sequential OFT sessions on days 7, 21, and 42 after PBS or EV treatment, using the naive group as control for evaluating memory of habituation to novelty. All groups presented normal short-term memory (evaluated only in the first session). Only the ISC group presented impairment of long-term memory (evaluated by comparing the first with the third session), an effect reversed with all EV treatment.

### 2.6. Novel Object Recognition Task (NORT)

Figure 7 shows that the same animals were subjected to all three sequential sessions on days 7, 21, and 42 after PBS or EV (EV0, EV1, and EV2) treatment, using the naive group as control. Stroke impaired both short- and long-term NORT memory, and effects abolished by EV treatment.

### 2.7. Elevated Plus-Maze Task (EPMT)

The task was performed only on the seventh day after treatment with EV0 or EV1. ISC animals spent less time in the open arms compared to the other groups, indicating a stroke-induced anxiogenic-like effect, which was abolished by EV0 and EV1 treatment (Figure 8).

### 2.8. Brain Angiogenesis

The number of branches and total length of blood vessels were evaluated in M2 and SS brain regions through images acquired in three positions (+2.20, +0.2, and −1.88 mm A.P. to Bregma) (Figure 9a).

Figure 9b,c show the number of branches and total length of blood vessels specifically in M2-ipsi- (I) and M2-contralateral (C) regions. Stroke decreased the number of branches and total length in both regions; however, EV2 treatment abolished the decrease specifically in the M2-I region. Figure 9d–i shows representative images of blood vessels in the M2-I and M2-C regions. In the SS regions, there was no difference of blood vessel parameters among the groups (See Appendix A).

## 3. Discussion

Since ischemic stroke is a significant cause of morbidity and mortality [1], research to develop new therapeutic strategies is constantly needed [15,20,47,48]. Scientific data from stroke patients have shown that behavioral and motor impairments are dependent on the infarct core and the penumbra zone in which reversible cell damage may be recovered by endogenous brain reactivity. The intensity and speed of the penumbra zone recovery may determine the size of the core and functional recovery after stroke [8]. Thus, innovative experimental therapeutic proposals, including the administration of EVs derived from MSCs, target this peri-infarct region, aiming to improve the recovery process [49] and, consequently, functional recovery [40]. Our research groups, and others, have already shown that, in the ischemic stroke rat model used here, the core and peri-infarct regions are located in the prefrontal cortex and hippocampus [17,18,19,20,21,50], which are brain structures involved in neuromotor and behavioral performance [51] evaluated in this study.

EVs are important mediators of communication between cells, and it has been identified that native brain cell-released EVs present tropism to injured regions [52,53,54]. Additionally, it has already been shown that intranasally administrated MSC-derived EVs were identified in brain cells [41,42], presenting tropism to injured brain regions [49]. In this study, we demonstrated that a single dose of EVs derived from hAT-MSCs intranasally administrated 24 h after brain insult resulted in a higher number of EVs in the peri-infarct regions, resulting in a decrease in infarct volume, alteration of the BBB permeability, and new vascularization. Notably, EVs administration reversed the impairments caused by brain insult on front paws symmetry, short- and long-term memory in OFT and NORT, and anxiety-like behavior. Thus, EV treatment contributed to the recovery of the peri-infarct brain region and simultaneously reversed the impairment in motor and behavioral performance, pointing to a potential role of hAT-MSC-derived EVs in functional recovery after stroke.

Stimulation of angiogenesis, as observed by EV treatment, has been shown to improve neurological and motor function in animal stroke models [55], an effect currently acknowledged as an outcome of EV transfer of proteins, mRNAs, and miRNAs to endothelial cells [33,56] and regulating protein expression [57]. Additionally, BBB impairment in ischemic stroke, as observed here, has also been documented [58,59,60], but its involvement in EVs therapeutic strategies, as indicated in this study, has not been previously reported.

Here, we demonstrate that intranasal hAT-MSC-derived EVs administered 24 h after stroke promoted long-term neuroprotective effects, offering a remarkably therapeutic window. Although the three EVs show protective properties, findings are not comparable since EV0, EV1, and EV2 are from different sources (2 healthy individuals and commercial hAT-MSCs). Together, these findings indicate that hAT-MSC-derived EVs are a promising potential therapeutic strategy for patients with focal permanent ischemic stroke.

## 4. Materials and Methods

### 4.1. hAT-MSCs: Sources, Culture and Characterization

The cells were obtained from commercial and human sources.

#### 4.1.1. Commercial hAT-MSCs

Cells were obtained from POIETICS Bank Adipose-Derived Stem Cells (cat. #PT-5006, donor 34464, male 58-year-old). These cells were negative for mycoplasma, bacteria, yeast, and fungi, as reported by the supplier company. A frozen vial containing ~1 × 10^6^ cells was thawed at 37 °C and plated in a 25 cm^2^ flask (TPP). Cells were cultured in Dulbecco’s modified Eagle’s medium (DMEM) (Sigma-Aldrich, San Luis, MI, USA) containing 10% fetal bovine serum (FBS) (Cripion, Andradina, São Paulo, BR), 100 units/mL penicillin, 100 μg/mL streptomycin (Gibco/Thermo Fisher Scientific, Waltham, MA, USA), 50 mg/L gentamicin (Sigma-Aldrich), and 2.5 mg/L fungizone (Sigma-Aldrich) (pH 7.4). After 24 h, debris and non-adherent cells were gently removed [36]. When adherent cells reached 80% confluence (passage1: P1), hAT-MSCs were detached with 0.25% trypsin/1 mM ethylenediaminetetraacetic acid (EDTA) (Sigma-Aldrich) and plated in flasks at a density of 1.5 × 10^4^ cells/75 cm^2^ (passage 2: P2). Cellular density was determined by manually counting the number of cells in each passage [61]. The cells were named as cell 0 (C0). C0 was expanded under the conditions described above and used only from passages four to eight [61,62,63].

#### 4.1.2. Patient-Derived hAT-MSCs

Cells were obtained from the subcutaneous adipose tissue of two (32-year-old and 34-year-old) women who underwent abdominal liposuction at the Hospital de Clínicas in Porto Alegre, RS, Brazil. Informed consent was obtained from all subjects involved in the study. The study was conducted in accordance with the Declaration of Helsinki, and the protocol was approved by Research and Graduate Group (Grupo de Pesquisa e Pós-Graduação: GPPG 2018-0374) and Research Ethics Committee (Comitê de Ética em Pesquisa CAEE: 94521618.4.0000.5327) of the Experimental Research Center at Hospital de Clínicas de Porto Alegre. Fresh adipose tissue was washed with PBS buffer, minced, and digested for 1 h in 0.1% collagenase at 37 °C. The digestion process was stopped by the addition of DMEM containing 20% FBS (Cripion). The digested suspension was filtered through a 70 µm nylon mesh cell filter to retain tissue debris. The filtered suspension was centrifuged at 400× *g* for 5 min. The stromal vascular fraction (pellet) was resuspended in DMEM + 20% FBS (Cripion) medium and cultured in a culture flask containing 25 cm^2^ (TPP -Techno Plastic Products, Trasadingen, CH) at 37 °C in a humidified 5% CO_2_ atmosphere. After 24 h, non-adherent cells were gently removed [61]. When adherent cells reached 80% confluence (passage 0: P0), confluent cells (hAT-MSCs) were detached with 0.25% trypsin/1 mM ethylenediaminetetraacetic acid (EDTA) (Sigma-Aldrich) and plated in flasks at a density of 1.5 × 10^4^ cells/75 cm^2^ (TPP) (passage1: P1). Cells were then cultured in DMEM (Sigma-Aldrich) containing 10% FBS (Cripion), 100 units/mL penicillin (Gibco/Thermo Fisher Scientific), 100 μg/mL streptomycin (Gibco/Thermo Fisher Scientific), gentamicin 50 mg/L (Sigma-Aldrich), fungizone (2.5 mg) (Sigma-Aldrich), and L-glutamine 4 mM (Gibco/Thermo Fisher Scientific). These cells were named cell 1—patient 1 (C1) and cell 2—patient 2 (C2). C1 and C2 were expanded under the same conditions described above and used only from the 4th to 8th passage [42,61]. C0, C1, and C2 hAT-MSCs were characterized by immunofluorescence using flow cytometry and confocal microscopy.

#### 4.1.3. Flow Cytometry

hAT-MSCs were centrifuged at 400× *g* for 5 min at room temperature, and the cell pellet was re-suspended in DMEM + 10% FBS and counted in a Neubauer chamber. Briefly, the cells were incubated with antibodies at a concentration of 1:50 for 4 h at 37 °C. The cell suspensions were centrifuged at 400× *g* for 5 min at room temperature, and cell pellets were resuspended in 200 μL of PBS. Ten thousand events were analyzed using flow cytometry (FACSCalibur™—BD Biosciense, Franklin Lakes, NJ, USA) [44]. Cells in passage 4 (P4) were characterized as hAT-MSCs by the presence of CD: CD34 (FITC conjugate mouse anti-human) (BD Biosciense), CD45 (FITC conjugate mouse anti-human) (Invitrogen, Waltham, MA, USA), CD90 (P.E. conjugate mouse anti-human) (BD Biosciense), and CD105 (R-PE conjugate mouse anti-human) (Invitrogen).

#### 4.1.4. Confocal Microscopy

An aliquot of 1 × 10^4^ hAT-MSCs was placed on a slide and analyzed by immunofluorescence. Cells were maintained under culture conditions for 72 h to allow adherence to coverslips. Cells were then incubated for 4 h at 37 °C with the same antibodies used for cytometry: CD34, CD45, CD90, and CD105, at a ratio of 1:500. The negative control was prepared by incubating only the secondary antibodies, Alexa Fluor 555 and 488 (Invitrogen). Cells on coverslips were gently washed with PBS (four times) to remove excess antibodies and then fixed in 4% PFA for 2 h. Following fixation, the cells were gently washed again with PBS and then fixed with Fluoromount (Sigma-Aldrich) onto a slide for further analysis. Images were acquired using an 8-bit grayscale confocal laser scanning microscope (Olympus FV1000, Shinjuku, Tokyo, Japan). Approximately 10 × 15 sections with 0.7 μm thick confocal were captured parallel to the coverslip (XY sections) using a ×20 objective. Z-stack reconstruction and analysis were conducted using ImageJ software (http://rsb.info.nih.gov/ij/, accessed on June 2018).

### 4.2. Extracellular Vesicles (EVs)

#### 4.2.1. EV Isolation and Purification

As cultured hAT-MSCs (P4-P8) reached 80% confluence, DMEM + 10% FBS medium was replaced by DMEM FBS-free medium to avoid EV contamination with FBS-proteins and FBS-EVs, following previous protocols [62,64,65,66]. After 72 h of culture, the medium was collected for EV isolation. The remaining cultured cells were supplemented with DMEM + 10% FBS to continue the culture [62].

For EV isolation, the collected medium was centrifuged (3 times) at 4 °C: (400× *g* for 15 min, then 2000× *g* for 15 min, and then 10,000× *g* for 30 min). The supernatants were filtered through a 0.22 μm membrane. The isolation was completed by centrifugation (100,000× *g* at 4 °C for 2 h). The supernatant was discarded, PBS was used to wash the pellet containing EVs, and the cell suspension was centrifuged at 100,000× *g* at 4 °C for 2 h. This centrifugation protocol (speed and time) was shown to preserve EVs in the pellet while preventing contamination with stress-related biological constituents [46,65,66,67]. Finally, the pellet was resuspended in 100 µL of PBS and stored at −20 °C [67]. Protein content was measured using a bicinchoninic acid (BCA) assay (Thermo Fisher Scientific) [61]. The vesicles isolated from C0, C1, and C2 cells were named EV0, EV1, and EV2, respectively.

#### 4.2.2. EVs Characterization

EVs were characterized by flow cytometry through the identification of membrane proteins CD63 and CD81 [45,46,65,66,67]. First, EVs were incubated with magnetic beads (Thermo Fisher Scientific, Invitrogen™), coated with primary antibody CD63 (Thermo Fisher Scientific) and CD81 (Thermo Fisher Scientific) for 18 h at 4 °C under gentle stirring. For each measurement, 10 µL of 1 mg/mL EV suspension was applied. To remove excess beads, immediately after incubation, EVs were washed with PBS, 2 mL of PBS was added for 5 min, and then the tube was placed in a magnet (to remove the beads) for 1 min, and the supernatant was discarded. Then, CD63, clone: MEM-259 (Invitrogen) and CD81, clone JS-81 (BD Pharmingen™, San Diego, CA, USA) antibodies (without granules) were added to the solution containing the EVs + magnetic beads. After 1 h of incubation, the EVs were gently washed by placing the tube on a magnet for 1 min, and the supernatant was discarded. We added 2 mL of PBS (to remove excess antibody) for 5 min and again placed the tube on a magnet for 1 min and discarded the supernatant. Finally, the EVs were resuspended in 200 μL PBS for analysis. Ten thousand events were analyzed by flow cytometry.

We used photon correlation spectroscopy to measure the particle size and polydispersity index (PDI). The EV suspension derived from hAT-MSCs (50 µL) at 1 mg/mL was diluted in 1 mL of PBS. All analyses were performed in triplicate using a Malvern Nano-ZS90^®^ (Malvern Instruments, Marvin City, UK) at 25 °C.

#### 4.2.3. EVs Measurement

Transmission electron microscopy (TEM) analysis, using a direct examination technique, was used to evaluate the diameter of EVs [49]. EV suspension (10 µL) and 1 mg/mL of protein were pipetted in aliquots into a grid covered with a carbon film (formvar/carbon) and dried at room temperature. Uranyl (Merck KGaA, Darmstadt, DE, USA) was used as a contrast agent. The sample was analyzed by TEM at 120 Kv in Microscopy and Microanalysis Center—UFRGS (JEM 1200 Exll-JEOL, Tokyo, Japan).

#### 4.2.4. EVs Labeling

EVs were labeled with the red fluorescent membrane dye PKH26 (Sigma-Aldrich). In brief, the EV-containing PBS solution was centrifuged at 100,000× *g* for 2 h at 4 °C, and the pellet was suspended with the diluent of the fluorescent kit. Filtered PKH26 (4 mM) and EVs (200 μg/mL) were mixed at a ratio of 1:1 for 5 min, followed by the addition of 5% BSA. To remove excess dye, the EVs were washed three times, and then 5 mL of PBS was added and centrifuged at 100,000× *g* for 2 h at 4 °C and the supernatant discarded. In the last centrifugation step, the stained EV pellet was suspended in PBS (0.5 mL). The solution was filtered through a 0.2 μm membrane filter to remove dye aggregates [42].

### 4.3. In Vivo Experiments

#### 4.3.1. Animals

Adult (90–120 days old) male Wistar rats weighing 350–400 g were maintained under controlled light (12/12 h light/dark cycle) at 22 °C ± 2 °C, with water and food ad libitum. All procedures were performed in accordance with the Guide for the Care and Use of Laboratory Animals and the Brazilian Society for Neuroscience and Behavior recommendations for animal studies. This study was approved by the Ethics Committee for the Use of Animals at the Universidade Federal do Rio Grande do Sul (project identification: 31888). A schematic of the procedure is shown in Figure 10.

Focal permanent ischemia and sham procedures: Anesthetized animals (under ketamine hydrochloride: 90 mg/kg i.p. and xylazine hydrochloride:10 mg/kg i.p.) were placed into a stereotaxic apparatus. After skin incision, the skull was exposed, and a craniotomy was performed by exposing the left frontoparietal cortex (+2 to −6 mm A.P. and −2 to −4 mm M.L. from the bregma). A focal permanent ischemic lesion was induced by thermocoagulation of the motor and sensorimotor pial vessels. Blood vessels were thermocoagulated by placing a hot probe near the dura mater for 2 min until a red-brown color indicated complete thermo-coagulation. Soon after, the skin was sutured, and the animals were placed on a heating pad at 37 °C until full recovery from anesthesia. Animals from the sham group were subjected to craniotomy as described above. Animals were randomly allocated to three treatment groups: sham, ischemic (ISC), and ischemic treated with EVs (ISC + EV). Our stroke model of focal cerebral permanent ischemia represents a protocol to investigate brain lesions with good reproducibility and low mortality [17,18,19,20,21,50,68,69].

#### 4.3.2. Intranasal EV Treatment

Intranasal EV treatment was performed 24 h after the ischemic or sham procedure. Sedated animals (O_2_ flow rate at 0.8–1.0 L/min with isoflurane levels of 2.5–3.0%) received a single 50 µL of EVs (ISC + EVs) or 50 µL PBS (naive, sham, ISC), slowly administrated for 30 s in both nasal cavities, as previously reported [20,70,71]. The 200 μg/kg EV dose was selected based on a dose–effect curve (ISC + PBS, ISC + 100 μg/kg, ISC + 200 μg/kg, and ISC + 300 μg/kg).

### 4.4. Brain Analysis

#### 4.4.1. Extracellular Fluorescent Vesicle (EV) Detection in the Rat Brain

The distribution of fluorescent EVs (PKH26- Sigma-Aldrich) in rat brains (naive and ISC + EV1) was analyzed 18 h after intranasal administration, as previously reported [42,49]. Anesthetized animals (ketamine hydrochloride: 90 mg/kg, 450 µL/kg i.p. and xylazine hydrochloride: 10 mg/kg, 300 µL/kg i.p.) were transcardially perfused with PBS using a peristaltic pump, followed by perfusion with 4% PFA (both 10 mL/min, 100 mL). Immediately, brains were dissected, immersed in 4% PFA (pH 7.4), and stored for a maximum of 7 days at 4 °C. Coronal brain sections (20 μm thick) were obtained using a vibratome (Leica, Wetzlar, Germany) at +2.20, 0,20, and −1,88 mm of Bregma (PAXINUS online Rat Brain Atlas: http://labs.gaidi.ca/rat-brain-atlas/ (accessed on January 2020). Brain slices were incubated for 5 min in the dark with 1 μg/mL Hoechst 33342 solution dye (Sigma-Aldrich) to detect cell nuclei. The slices were washed with PBS (four times), and the slices were fixed with the fluoro mount (Sigma-Aldrich). Slice images for counting EVs were acquired using an 8-bit grayscale confocal laser scanning microscope (Olympus FV1000). Approximately 10–15 sections with 0.7 μm thick confocal were captured parallel to the coverslip (XY sections) using a ×60 objective. Z-stack reconstructions and analyses to count the vesicles in the brain tissue were conducted using ImageJ Free software (National Institutes of Health, Bethesda, MD, USA), and background noise was removed using the “subtract background” tool. Images were converted to binary masks using the default threshold option, and vesicles were counted with the “analyze particles” tool (size = 0.05–0.90 μm). These settings were programmed into a macro and used for all the analyzed images (http://rsb.info.nih.gov/ij/ (accessed on June 2018)).

#### 4.4.2. Short-Term Evaluation of Infarct Volume

For evaluating infarct volume, naive, sham, ISC, ISC + EV0, and ISC + EV2 animals were sedated 48 h after treatment (O_2_ flow rate of 0.8–1.0 Â mL/min with isoflurane levels of 2.5–3.0%), decapitated, and the brain removed. Coronal sections of the whole brain were sliced at 2 mm, and slices were immersed in 2% 2, 3, 5-Triphenyl-tetrazolium chloride (TTC) (Sigma-Aldrich) for 30 min at 37 °C. After incubation, the slices were dipped in 4% buffered paraformaldehyde (pH 7.4) for 24 h. The infarct area was evaluated as an area devoid of red staining. The infarct volume was measured using the ImageJ software [21].

#### 4.4.3. Brain Angiogenesis

After 42 days of treatment, animals from the naive, ISC, and ISC + EV2 groups were anesthetized and received intracardiac injection of 50 mg/mL (500 µL) fluorescein isothiocyanate-dextran amine (Merck KGaA) to label blood vessels. After 5 min, rat brains were excised, immediately fixed in 4% PFA, and cut into 30 µm coronal slices in a vibratome. Images were acquired using a fluorescence microscope (Nikon, Tokyo, Japan). The images were taken from the ipsilateral and contralateral sides in the secondary motor cortex (M2) and somatosensory (SS) regions using the following coordinates: +2.20, 0.2, and −1.88 mm A.P. to Bregma (PAXINUS online Rat Brain Atlas: http://labs.gaidi.ca/rat-brain-atlas/ (accessed on January 2020). Blood vessel parameters, such as the total length (sum of the length of segments and isolated elements) and the number of branches, were quantified using the Angiogenesis Analyzer Plugin (Gilles Carpentier Research) ImageJ software [72] (https://imagej.nih.gov/ij/ accessed on June 2019) [73,74].

#### 4.4.4. BBB Permeability

Evans blue penetration into brain parenchyma: Naive, sham, ISC, and ISC + EV2 animals were anesthetized (ketamine hydrochloride: 90 mg/kg, 450 µL/kg i.p. and xylazine hydrochloride 10 mg/kg, 300 µL/kg i.p.) 48 h after treatment, and 3 mL/kg of 2% Evans blue (EB) (Sigma-Aldrich, San Luis, MI, USA) solution in saline was administered through the gingival artery (Appendix A). After 1 h, the animals were subjected to cardiac perfusion using a peristaltic pump (10 mL/min, with PBS, 100 mL). The animals were decapitated, and whole brain images were obtained. The brains were then sliced (coronal sections) at 2 mm and images obtained. The slices from each brain were macerated and homogenized in 2.5 mL of PBS and vortexed for 2 min to measure the amount of EB in the brain parenchyma. For protein precipitation, 2.5 mL of 50% trichloroacetic acid was added to the homogenate, incubated for 12 h at 50 °C, and centrifuged at 14,000× *g* for 10 min. The concentration of the blue color was measured in the supernatant using a spectrophotometer at a wavelength of 620 nm. EB dye was expressed in micrograms per gram of brain tissue [75,76].

CSF albumin levels: Albumin assay was performed using high-performance liquid chromatography coupled to a fluorescence detector (HPLC-FLD). The CSF method was validated according to FDA guidelines [77,78]. HPLC-FLD consisted of an L.C. Shimadzu system (Kyoto, Japan) equipped with a 20AT pump, a DGU-14A degasser, a thermostat for a CTO-10A column, and a fluorescence detector, R.F. 20A. Data acquisition and processing were performed using the L.C software (LabSolution software, Shimadzu, Korneuburg, Niederösterreich, AT, USA) version 5.73. The FLD was set at 278 nm (excitation) and 335 nm (emission). An Agilent reversed-phase ZORBAX SB-C18 column (5 μm particle size, 250 mm × 4.6 mm) was used. The method was performed using the following gradient conditions: solvent A (H_2_O + 0.1% formic acid) and solvent B (acetonitrile (ACN) as follows: A → 65% B → 35% (0–5 min), A → 70% B → 30% (5–10 min), A → 65% B → 35% (10–17 min). The flow rate was set at 0.7 mL/min. Sample preparation was performed by adding 10 µL of liquor to 40 µL of ACN and mixing in a vortex. The solution was transferred to conical vials and 10 µL was injected. Albumin stock solutions 1 mg/mL in water were stored at −20 ± 2 °C. For each day of analysis, standard solutions of albumin were prepared at 0.1, 0.5, 1, 10, 50, and 100 µg/mL.

### 4.5. Behavioral Tasks

A group of animals was submitted to the cylinder task, open field task, and novel object recognition task; another group of animals was submitted to the cylinder task and elevated plus-maze task. The cylinder task, which evaluates neurologic dysfunction (front paws symmetry), was performed before surgery; the presence of ~100% basal symmetry was used as inclusion criterion in the study.

#### 4.5.1. Cylinder Task (CT)

The CT can evaluate the motor symmetry of the front paws. The apparatus consists of a transparent glass cylinder 20 cm in diameter and 30 cm in height (20 raising movements were counted). All animals (naive, naïve + EV0, sham, ISC, and ISC + EVs groups) were firstly submitted to this task 24 h before surgery to verify basal symmetry. This task evaluated how the rats raised their bodies in contact with their paws on the cylinder wall. The ipsilateral (to the lesion), contralateral, or both front paws preferences were counted in a blinded analysis. The asymmetry of each animal was calculated using the following formula: asymmetry = (% of ipsilateral use = ipsilateral paw use/sum ipsilateral + contralateral + use of both paws) − (% of contralateral paw use/sum of ipsilateral and contralateral paws). The performance was recorded using ANY-Maze software version 6.3 (Stoelting Co., Wood Dale, IL, USA). The asymmetry percentage was converted into a symmetry percentage [50]. The same group of animals were submitted to CT 24 h before surgery and on the 3rd, 7th, 14th, 21st, 28th, 35th, and 42nd day after EV treatment. At the end of each task, the apparatus was cleaned using 70% ethanol solution.

#### 4.5.2. Open Field Task (OFT)

This task evaluates habituation to novelty (assessing short- and long-term memory through exploratory activity) and locomotor activity [79,80]. The arena consisted of a black cage measuring 50 × 50 × 50 cm. The individual sessions lasted 10 min. The animals (naive, naïve + EV0, ISC, and ISC + EVs groups) were submitted to the task on days 7, 21, and 42 after EV treatment. Short-term memory was evaluated considering the decrease in locomotion during the first 5 min of the 1st session (only on the 7th day). Long-term memory was evaluated considering the decrease in locomotion during the first minute through successive sessions (from the 1st to the 3rd session). At the end of each session, the apparatus was cleaned with 70% ethanol solution. The task was recorded and analyzed using ANY-Maze software.

#### 4.5.3. Novel Object Recognition Task (NORT)

Behavioral sessions lasting 10 min were performed on days 7, 21, and 42 after EV treatment. At 90 min after the OFT session, object recognition (OR) short- and long-term memories were evaluated [81]. The animals were individually placed on the periphery of the arena for further exploration. Two identical familiar objects (FOs) were placed in the arena, and the animals could explore them for 10 min (training session). Sniffing and touching objects were considered exploratory behaviors. The animals were then removed from the arena, and 90 min after the training session, each animal was placed back into the arena to evaluate short-term memory (first test session). One of the two FOs used in the training session was replaced by a new distinct object (NO). Long-term memory was evaluated 24 h after the training session, when the animals were placed back in the arena with the same FO used in the training session and the first test session (short-term memory); however, the same NO was displaced to a different position. In all sessions, the time spent exploring the objects was recorded using ANY-maze software. The results are expressed as a percentage of the time spent exploring each object. Animals that recognized the novel object (short-term memory) or its new position (long-term memory) explored more than 50% of the total exploration time of both objects. At the end of each session, the apparatus was cleaned using 70% ethanol solution.

#### 4.5.4. Elevated Plus-Maze Task (EPMT)

This task is widely used to study anxiety-like behavior [82]. The apparatus had two open arms (50 cm long × 10 cm wide) and two closed arms (50 cm long × 10 cm wide × 40 cm high), separated by a square central platform (5 × 5 cm). The apparatus was placed 70 cm above the floor. The animals (naive, naïve + EV0, ISC, ISC + EV0, and ISC + EV1) were habituated in a red-light room for 1 h before starting the task. The percentage of time spent in the open and closed arms was assessed. Anxiety-like behavior was considered as the increase of time spent in the closed arms. Each animal was submitted to this task once on the 7th day after treatment with EVs. ANY-maze software was used to record behavioral performance for 5 min. For baseline, we used the results of naive group performance. At the end of each session, the equipment was cleaned with 70% alcohol.

### 4.6. Statistical Analyses

Brain lesion size and the number of brain vesicles showed a non-homogeneous distribution (Shapiro–Wilk test *p* < 0.05); thus, statistical analyses of both experiments were performed using the Mann–Whitney test, and the data were reported as median and interquartile range.

The results of BBB integrity, angiogenesis, OFT short-term memory, and EPMT were evaluated using unpaired *t* tests. Two-way RM ANOVA was applied for CT, followed by Sidak’s multiple comparison test. OFT long-term memory was evaluated by two-way ANOVA, followed by Sidak’s multiple comparison test. Unpaired t tests were used for the NORT, with a theoretical average of 50%. Comparison between EVs (EV0, EV1, and EV2) were not performed due to differences in EVs sources (2 donors and one commercial). In the parametric distribution, the data were presented as mean ± SD. All analyses were performed using GraphPad Prism version 6.0 (San Diego, CA, USA).

## Figures and Tables

**Figure 1 ijms-22-12860-f001:**
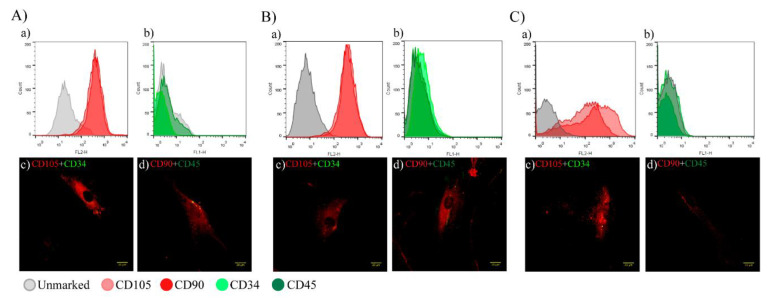
Characterization of hAT-MSCs. (**A**–**C**) Images of flow cytometry and fluorescence microscopy of CD markers specific to hAT-MSCs (**A** = C0, **B** = C1, and **C** = C2). (**A****a**,**Ba**,**Ca**) CD105 and CD90 (positive in 70% of the cells). (**A****b**,**Bb**,**Cb**) CD34 and CD45 (unexpressed). The positive result was a peak fluorescence shift. These results were compared with C0 unstained (gray). Representative images of fluorescence microscopy, 40× objective: (**Ac**,**Bc**,**Cc**) Merged CD105 (red) and CD34 (green). (**Ad**,**Bd**,**Cd)** Merged CD90 (red) and CD45 (green). n = 3; scale bars = 20 µm.

**Figure 2 ijms-22-12860-f002:**
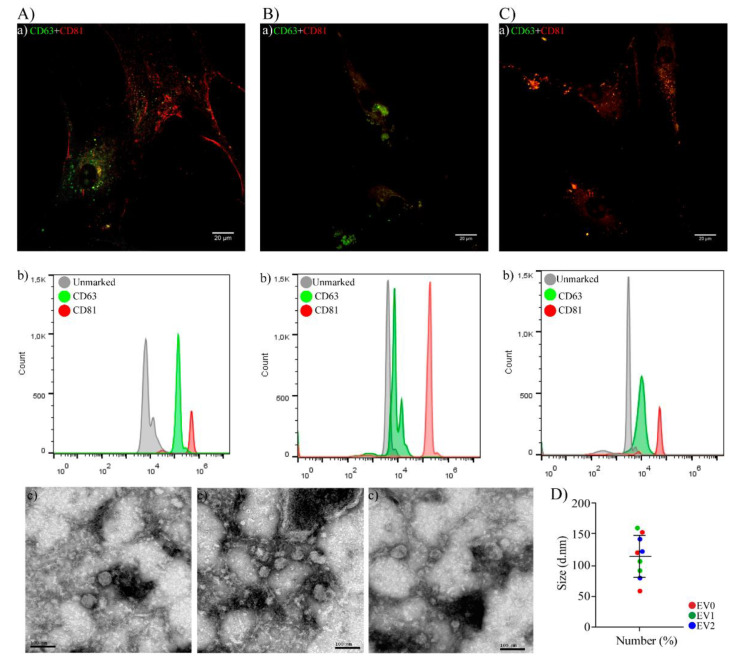
Characterization of EVs. The fluorescence signals of the following EV markers were observed inside the cells, EV0: (**Aa**), EV1: *(***Ba**), and EV2: (**Ca**) Merged CD81 (red) and CD63 (green). Scale bars: 20 µm. Released EV characterization by flow cytometry with specific CD markers. The positive result was a peak fluorescence shift for CD63 (green) and CD81 (red). These results were compared with EV0 incubated only with beads (gray), EV0: (**Ab**), EV1: (**Bb**), and EV2: (**Cb**). Transmission electron microscopy (TEM) image by direct examination, demonstrating the diameter of hAT-MSC-derived EV0: (**Ac**), EV1: (**Bc**), and EV2 (**Cc**), by MET JEM 1200 EXll, magnification 200k, scale bar = 100 nm. (**D**) Average diameter (140 nm) evaluated by the Zetasizer instrument: EV0 (red), EV1 (green), and EV2 (blue).

**Figure 3 ijms-22-12860-f003:**
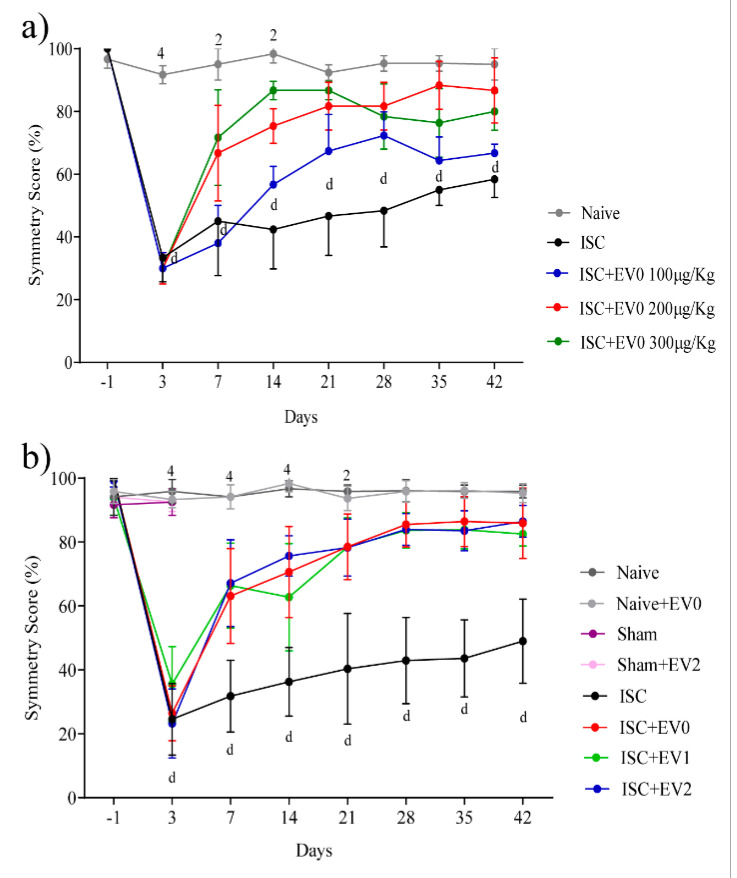
(**a**) EV0 dose curve for symmetry score in cylinder task (CT). Naive, ISC, ISC + EV0 (100, 200, or 300 μg/kg), n = 3/group. Day-1 refers to baseline symmetry, evaluated 24 h before induction of stroke. Data are expressed as mean ± SD and analyzed by two-way ANOVA followed by Tukey’s test: ^d^
*p* < 0.0001 compared ISC and ISC + EV0 100 µg/Kg to naive; ^2^
*p* < 0.01, ^4^
*p* < 0.0001, compared naive to ISC + EV0 200 and 300 µg/kg groups. (**b**) EVs (200 μg/kg) effect on symmetry score. Naive (*n* = 8), naive + EV0 (n = 6), sham (n = 11), sham + EV2 (n = 5), ISC (n = 22), ISC + EV0 (n = 17); ISC + EV1 (n = 17); ISC + EV2 (n = 17). Data are expressed as means ± SD, analyzed by two-way ANOVA followed by Tukey’s multiple comparisons test; ^d^
*p* < 0.0001 compared ISC to naive; ^2^
*p* < 0.01, ^4^
*p* < 0.0001 naive group compared to ISC + EV groups.

**Figure 4 ijms-22-12860-f004:**
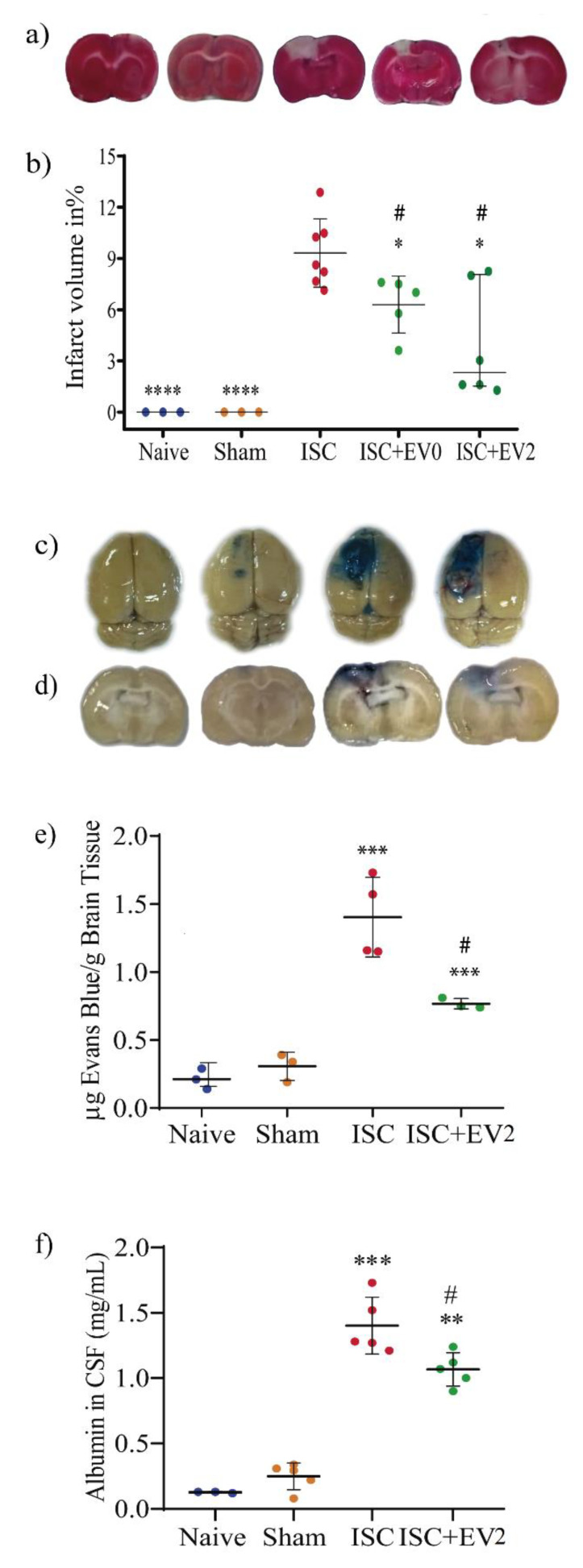
(**a**) Representative images of infarct volume (TTC staining). (**b**) EV0 and EV2 treatment significantly decreased infarct volume. Based on the Shapiro–Wilk test, we used the Mann–Whitney test of the unpaired t test. Data are reported as median with interquartile range * *p* < 0.05 and **** *p* < 0.0001 compared to ISC and ^#^
*p* < 0.05 compared to naive and sham groups. (**c**,**d**) Representative images of Evans blue penetration in whole brain (**c**) and in brain slice 0 mm to Bregma (**d**). (**e**) Quantification of Evans blue in the brain. Statistical analysis using unpaired t test. Data are reported as mean ± SD. *** *p* < 0.001 compared to naive and sham groups, ^#^
*p* < 0.05 compared to ISC group. (**f**) Albumin levels in CSF. Statistical analysis using unpaired *t* test. Data are reported as the mean ± SD. ** *p* < 0.01 and *** *p* < 0.001 compared to naive and sham groups, ^#^
*p* < 0.05 compared to the ISC group.

**Figure 5 ijms-22-12860-f005:**
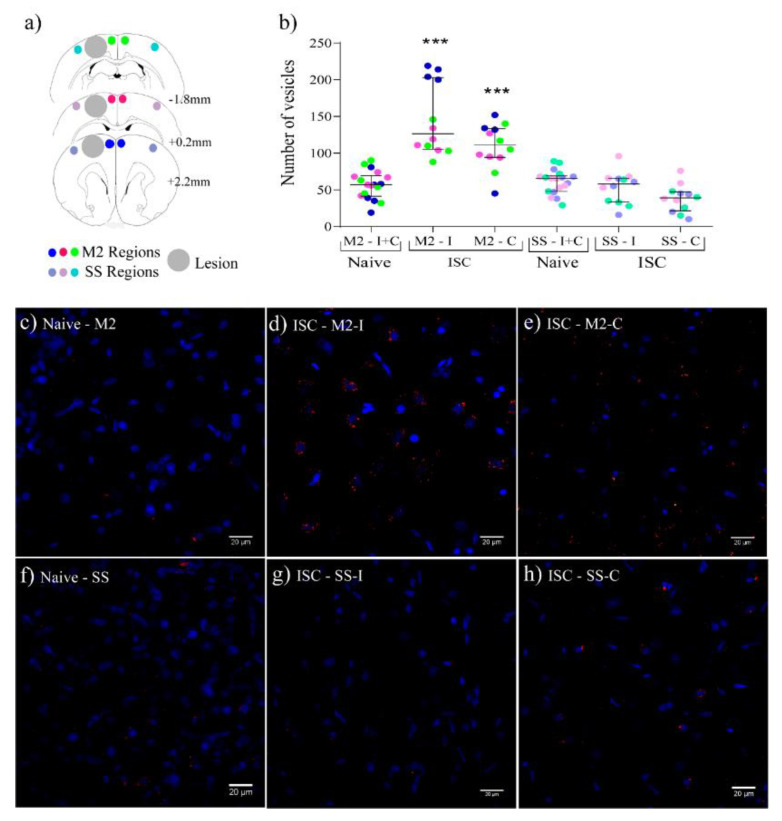
EV0 distribution in the brain measured 18 h after treatment. (**a**) Schema of the regions where the images were taken. Supplementary motor cortex (M2) and somatosensory (SS) (positions +2.20, 0.2, and −1.88 mm to Bregma). Big gray cycle: core; small color cycles: analyzed sub regions. (**b**) Figure indicating the number of vesicles in the 3 positions (3 different colors) of ipsi- (I)- and contra (C)-lateral M2 and SS regions. (**c**–**h**) representative images of the M2 and SS regions at +0.20 mm to the Bregma: (**c**) naive M2, (**d**) ISC M2-I), (**e**) ISC M2-C, (**f**) naive SS, (**g**) ISC SS-I, (**h**) ISC SS-C. Statistical analysis by Mann–Whitney test. Data are reported as median and interquartile range. *** *p* < 0.001 compared to all other groups without asterisks (n = 3). Scale bars = 20 µm.

**Figure 6 ijms-22-12860-f006:**
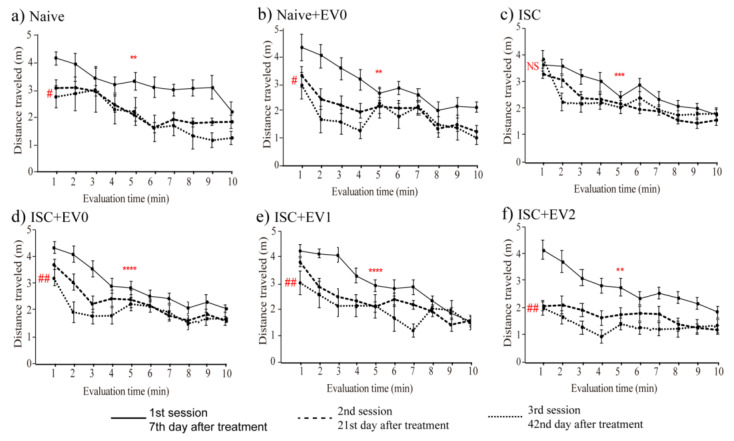
Open field task. Sessions were performed on days 7, 21, and 42 after treatment. Groups: (**a**) naive (n = 8), (**b**) naive + EV0 (n = 6), (**c**) ISC (n = 16), (**d**) ISC + EV0 (n = 16), (**e**) ISC + EV1 (n = 15), and (**f**) ISC + EV2 (n = 16). Statistical analysis using 2-way ANOVA, followed by the Tukey’s multiple comparisons test. Data are reported as mean ± SEM. ** *p* < 0.01, *** *p* < 0.001, **** *p* < 0.0001 comparing the 1st min with the 5th min only in the first session; ^#^
*p* < 0.05, ^##^
*p* < 0.01, comparing the 1st min of the first session with the 1st min of the second/third or third sessions. NS = not statistically significant.

**Figure 7 ijms-22-12860-f007:**
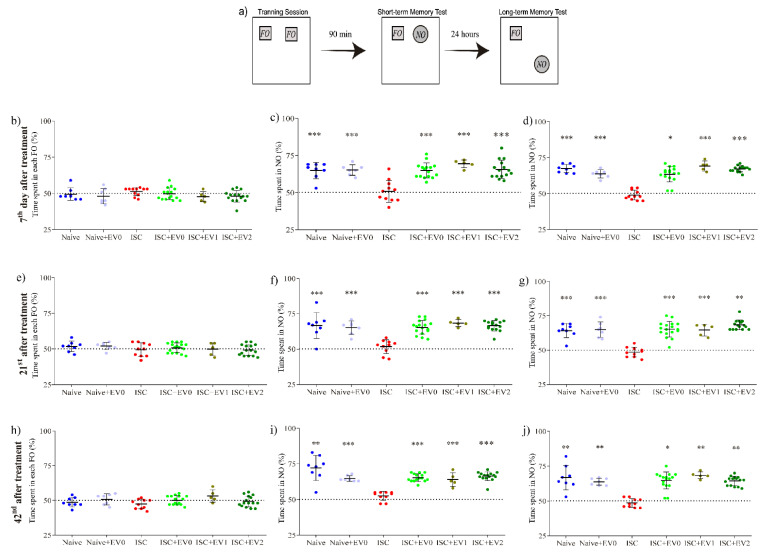
Novel object recognition task. Sessions were performed on days 7, 21, and 42 after treatment. (**a**) Scheme of the protocol using familiar object (FO) and novel object (NO). Groups: naive (n = 8), naive + EV0 (n = 7), ISC (n = 16), ISC + EV0 (n = 16), ISC + EV1 (n = 5), and ISC + EV2 (n = 16). (**b**,**e**,**h**) 3 successive training sessions; (**c**,**f**,**i**) 3 successive short-term memory test sessions; (**d**,**g**,**j**) 3 successive long-term memory test sessions. Unpaired t test with a theoretical average of 50%. Data reported as mean ± SD; * *p* < 0.05, ** *p* < 0.01, and *** *p* < 0.001, compared with the theoretical average of 50%.

**Figure 8 ijms-22-12860-f008:**
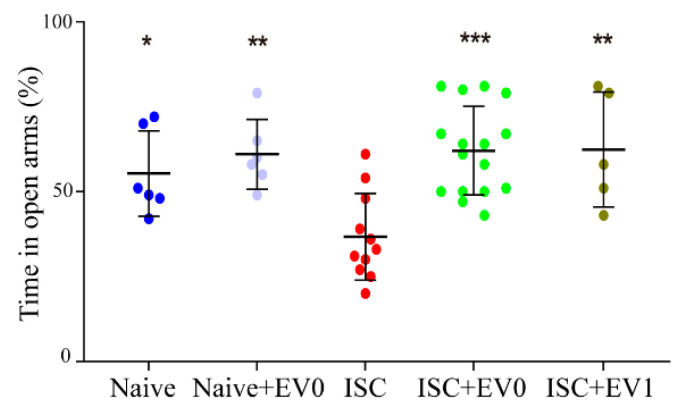
Elevated plus-maze task. Session was performed on day 7 after treatment. Groups: naive (n = 6), naive + EV0 (n = 6), ISC (n = 11), ISC + EV0 (n = 16), and ISC + EV1 (n = 5). Statistical analysis using the unpaired t test. Data are reported as mean ± SD, * *p* < 0.05, ** *p* < 0.01, *** *p* < 0.001, compared to ISC group.

**Figure 9 ijms-22-12860-f009:**
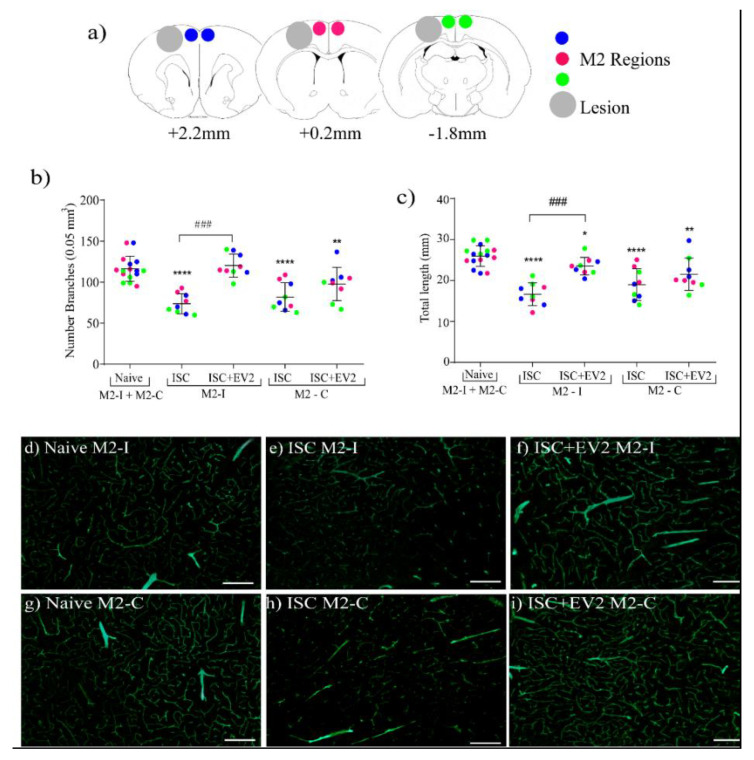
Brain blood vessel parameters measured on day 42 after treatment with EV2 in supplementary motor cortex region (M2); volume of piece: 0.05 mm^3^. (**a**) Scheme on brain regions analyzed: ipsi- peri-infarct region and its contralateral equivalent (positions +2.20, +0.2, and −1.88 mm to Bregma); (**b**) number of vessel branches in M2-ipsi- (M2-I) and M2-contralateral (M2-C) regions; (**c**) total vessel lengths in M2-I and M2-C regions; (**d**–**i**) representative images of blood vessel in M2 region located at position +0.20 mm to Bregma: (**d**) naive M2-I; (**e**) ISC M2-I; (**f**) ISC + EV2 M2-I; (**g**) naive M2-C; (**h**) ISC M2-C; (**i**) ISC + EV2 M2-C. Statistical analysis using the unpaired *t* test. Data are reported as mean ± SD. * *p* < 0.05, ** *p* < 0.01, **** *p* < 0.0001, compared to naive group; ^###^
*p* < 0.001 compared to ISC group (n = 3). Scale bars = 20 µm.

**Figure 10 ijms-22-12860-f010:**
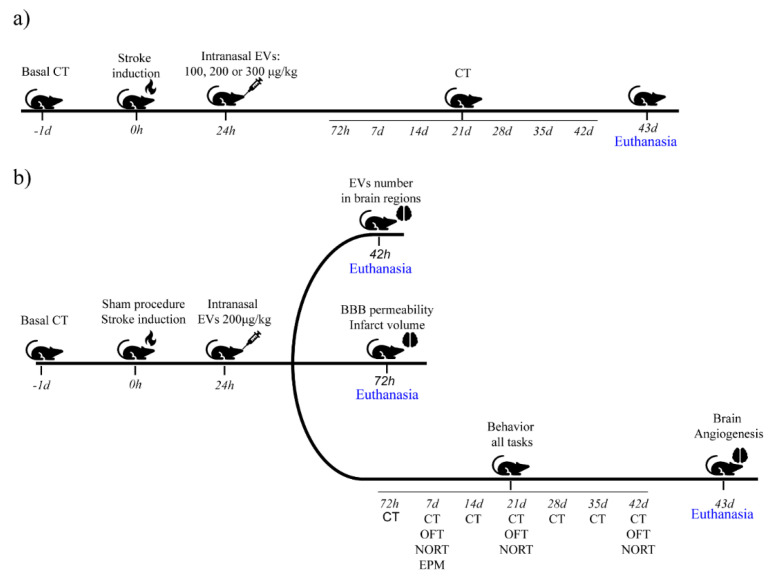
Experimental scheme. (**a**) Dose curve, animals received PBS or EVs (100, 200 or 300 µg/kg) intranasally 24 h after surgery. (**b**) Short- and long-term analyses: animals received PBS or EVs 200 µg/kg, intranasally 24 h after surgery. Short-term analyses: distribution of EVs in the brain, BBB permeability, and infarct volume; long-term analyses: CT: cylinder task; OFT: open field task; NORT: novel object recognition task; EPMT: elevated plus-maze task.

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
