# Peer review of "Functional Recovery Caused by Human Adipose Tissue Mesenchymal Stem Cell-Derived Extracellular Vesicles Administered 24 h after Stroke in Rats"

_ijms, 2021, doi:10.3390/ijms222312860_

Round 1
Reviewer 1 Report
1-The study entitled “Functional recovery caused by human adipose tissue mesenchymal stem cell-derived extracellular vesicles administered 24 hours after stroke in rats” approaches an interesting topic which is the use of intranasally administered EVs to treat stroke. It is well designed, conducted, and written. However, there is an article (Pathipati, Praneeti, et al 2021) that has already reported a similar approach. It should be emphasized in the introduction section what is novel about this study.
(Pathipati, Praneeti, et al. "Mesenchymal Stem Cell (MSC)–Derived Extracellular Vesicles Protect from Neonatal Stroke by Interacting with Microglial Cells." Neurotherapeutics (2021): 1-14)
2-EVs are classified by (Akers et al., 2013) into three main classes: (a) apoptotic bodies, 800–5,000 nm diameter, (b) microvesicles MVs, (50–1,000 nm diameter), and (c) exosomes, 40–150 nm diameter. It is not clear why exosomes in this study were indicated as EVs, not exosomes.
3-However it is understood how TEM was used to evaluate the diameter of EVs, it remains unclear how it was used to evaluate their purity.
4- Pathipati, Praneeti, et al 2021 showed that delayed intranasal administration of MSCs after induction of experimental stroke improved the long-term functional outcomes. Did you explore a delayed administration of EVs to compare between 24 h after stroke with delayed therapy?
5-The distribution of fluorescent EVs in rat brains was analyzed 18 h after intranasal administration. This will nicely investigate the delivery of EVs to the brain tissue a few hours after treatment (in the immediate term). Did you investigate the retention of such delivered EVs in the short- and long term? Maybe the same time you investigated brain Angiogenesis (6 weeks after treatment).
6- The study discussion has to elaborate more on discussing why AD-MSCs among all MSC types were chosen for this study and why that particular ischemia model was adopted.
Author Response
First of all, we want to thank you for reviewing our manuscript and we hope your responses are satisfactory.
In addition, we perform MS English correction as requested.
Response to Reviewer 1 Comments:
Point 1: The study entitled “Functional recovery caused by human adipose tissue mesenchymal stem cell-derived extracellular vesicles administered 24 hours after stroke in rats” approaches an interesting topic which is the use of intranasally administered EVs to treat stroke. It is well designed, conducted, and written. However, there is an article (Pathipati, Praneeti, et al 2021) that has already reported a similar approach. It should be emphasized in the introduction section what is novel about this study.
Response 1: Thank you so much for your observation; we included this reference in our MS: Pathipati, Praneeti, et al. "Mesenchymal Stem Cell (MSC)–Derived Extracellular Vesicles Protect from Neonatal Stroke by Interacting with Microglial Cells." Neurotherapeutics (2021): 1-14. This paper is very interesting and really contributes to the understanding of the EVs neuroprotective role against experimental stroke.
Our study presents significant differences from the Pathipati, et al (2021) paper. Below, we specify what we consider to be relevant distinct features between both studies.
- They used neonatal mice, postnatal day P9-P10 C57Bl6 wild type (WT) mice of both sexes. We used adult male Wistar rats (90-120 days).
- They used a focal ischemic transient stroke (internal carotid artery occlusion for 3h). We used a focal ischemic permanent stroke (termocoagulation of pial vessels).
- They used a mix of MSCs from long bones of 4- to 6-week old male C57Bl6 mice and microglia isolation from acutely injured neonatal brains. We used EVs from AD-MSC obtained from healthy individuals submitted to liposuction.
- Their EVs administration was via ICV (ipsilateral to the tMCAO) or via intranasal injection. We used only intranasal injection.
- They investigated the role of microglia on the EVs effect. We did not perform this approach.
- They used the impact on the size of the ipsilateral hemisphere (to tMCAO) and the neurochemical alterations as the neuroprotective EVs effect. We used in vivo alterations on neurological behavioral parameters, as well as the decrease of the infarct size as neuroprotective EVs effect.
Point 2: EVs are classified by (Akers et al., 2013) into three main classes: (a) apoptotic bodies, 800–5,000 nm diameter, (b) microvesicles MVs, (50–1,000 nm diameter), and (c) exosomes, 40–150 nm diameter. It is not clear why exosomes in this study were indicated as EVs, not exosomes.
Response 2: We choose the name of extracellular vesicles (EVs) taking into account the guidelines established by Clotilde Théry et al, 2018: Minimal information for studies of extracellular vesicles 2018 (MISEV2018): a position statement of the International Society for Extracellular Vesicles and update of the MISEV2014 guidelines:
"The International Society for Extracellular Vesicles, ISEV, endorses “extracellular vesicle” (EV) as the generic term for particles naturally released from the cell that are delimited by a lipid bilayer and cannot replicate, i.e. do not contain a functional nucleus. Since consensus has not yet emerged on specific markers of EV subtypes, such as endosome-origin “exosomes” and plasma membrane-derived “ectosomes” (microparticles/microvesicles) assigning an EV to a particular biogenesis pathway remains extraordinarily difficult unless, e.g. the EV is caught in the act of release by live imaging techniques."
Thus, we kindly ask the reviewer to agree with the use of EVs throughout the text. However, if the reviewer understands that the term “exosomes” is more appropriate, we will adapt the text.
Point 3: However, it is understood how TEM was used to evaluate the diameter of EVs, it remains unclear how it was used to evaluate their purity.
Response 3: Thank you for this observation. We used TEM to evaluate the diameter of EVs. As we did not observe other structures, organelles or cells, we considered this methodology adequate to evaluate the EVs purity. If the reviewer considers that TEM is not adequate to measure the purity, we will eliminate this observation.
Point 4: Pathipati, Praneeti, et al, 2021 showed that delayed intranasal administration of MSCs after induction of experimental stroke improved the long-term functional outcomes. Did you explore a delayed administration of EVs to compare between 24 h after stroke with delayed therapy?
Response 4: We observed that 200µg/kg EVs administration 48 hours after stroke caused less neurological recovery (front paws symmetry) than the administration 24 hours after stroke (please see figure attached). Thus, in this study the EVs administration protocol in all experiments was 200µg/kg 24 hours after stroke.
Fig. EV0 time curve for Symmetry Score in Cylinder Task (CT). ISC and ISC+EV0 24h (Blue) or ISC+EV0 48h (Red), 200 μg/kg, n = 3/group. Day -1 refers to baseline symmetry, evaluated 24 hours before induction of stroke. Data are expressed as mean ± SD and analyzed by two-way ANOVA followed by the Tukey’s test: ***p<0.001 and ****p<0.0001 compared to ISC+EV0 24h to ISC group.
Point 5: The distribution of fluorescent EVs in rat brains was analyzed 18 h after intranasal administration. This will nicely investigate the delivery of EVs to the brain tissue a few hours after treatment (in the immediate term). Did you investigate the retention of such delivered EVs in the short- and long term? Maybe the same time you investigated brain Angiogenesis (6 weeks after treatment).
Response 5: Thank you for the interesting observation. Aiming to correlate the EVs short-term neuroprotective effects (on infarct brain volume and on BBB permeability) with the entrance of EVs into the brain parenchyma, we evaluated their distribution in the brain 18 hours after treatment. It is important to emphasize that we consider that the EVs presence in the brain a few hours after intranasal administration could also be correlated with long-term neuroprotective effects of EVs.
We found two studies (PERETS, et al, 2019 DOI: 10.1021/acs.nanolett.8b04148 and XU, et al, 2020 DOI: 10.2147/IJN.S271519) that measured the long-term presence of EVs in brain parenchyma: 4 days after treatment and 14 days after treatment, respectively. Both showed that EVs remained in the brain tissue, but the amount of these EVs was not evaluated.
Thus, based on these two studies and on the relevant observation of the reviewer, we will consider in the next steps of our research to evaluate the long-term presence of EVs in brain parenchyma.
Point 6: The study discussion has to elaborate more on discussing why AD-MSCs among all MSC types were chosen for this study and why that particular ischemia model was adopted.
Response 6: There are various reports using AD-MSCs as a source of EVS. Here, in Brazil, to use human adipocyte tissue is more feasible because the tissue is collected through a minimally invasive procedure that can be obtained from healthy individuals. These characteristics contribute to the approval by ethic committees.
Our ischemic model was adopted due to the following reasons: 1) it is an ischemic focal permanent stroke model (~85% of total strokes in the world); 2) it presents high reproducibility of results; 3) it presents very low mortality rate; 4) we have many publications with this model.

Reviewer 2 Report
The manuscript entitled “Functional recovery caused by human adipose tissue mesenchymal stem cell-derived extracellular vesicles administered 24 hours after stroke in rats”. The authors have investigated the use of extracellular vesicles (EVs) that were derived from human adipose tissue derived mesenchymal stem cells (hAT-MSCs), which were either obtained commercially or obtained from 2 different patients. The authors have performed a battery of behavior tasks to show the behavior outcomes in the ischemic stroke rats following treatment with EVs. In addition, there is a good outcome in terms of brain angiogenesis following treatment in rats.
However, some enthusiasm was reduced because of major issues described below:
- The novelty of the study is difficult to grasp as multiple report already exist with similar topics. EVs and MSCs are being used as a potential treatment for stroke (https://stemcellres.biomedcentral.com/articles/10.1186/s13287-020-01601-1). The authors need to write clearly and compare their study to existing published articles.
- The timeline and figure 10 are not very clear. Please mention the animals and groups very clearly and explain the timeline from baseline behavior until euthanasia.
- Why EVs (especially the ones obtained from patients) were kept separately for treatment and not pooled in together. What if EVs obtained from one patient is more effective than the other patients? Why EVs were obtained only from 2 female patients? At least 3 patients are needed to have minimum randomization and different gender.
- Are the authors comparing the therapeutic efficacy of commercial EVs versus EVs from patients? If so, what is the rationale?
- When the animals were intranasally injected with EVs, was just one nostril used or were they injected into both the nostrils?
Overall, this study does not have any novelty described by the authors, the methods and animal grouping are not very clear, and the study is biased by having individual EVs obtained from one gender from only 2 patients. There is no randomization, therefore, this reviewer rejects this manuscript.
Author Response
Response to Reviewer 2 Comments:
The manuscript entitled “Functional recovery caused by human adipose tissue mesenchymal stem cell-derived extracellular vesicles administered 24 hours after stroke in rats”. The authors have investigated the use of extracellular vesicles (EVs) that were derived from human adipose tissue derived mesenchymal stem cells (hAT-MSCs), which were either obtained commercially or obtained from 2 different patients. The authors have performed a battery of behavior tasks to show the behavior outcomes in the ischemic stroke rats following treatment with EVs. In addition, there is a good outcome in terms of brain angiogenesis following treatment in rats.
However, some enthusiasm was reduced because of major issues described below:
Firstly, we would like to thank the reviewer for reading our MS and for making considerations and suggestions that will certainly help to improve our MS.
Below we respond to all the topics commented by the reviewer.
Point 1: The novelty of the study is difficult to grasp as multiple reports already exist with similar topics. EVs and MSCs are being used as a potential treatment for stroke (https://stemcellres.biomedcentral.com/articles/10.1186/s13287-020-01601-1). The authors need to write clearly and compare their study to existing published articles.
Response 1: We thank the reviewer for the comment. Although there are several articles that investigated the neuroprotective effect of EVs in experimental models of stroke, we believe that very few presented the set of methodologies and results shown in our MS. And we also trust that no previous work presented a global set as presented in this study.
We performed data collection on October 26, 2021, using the PubMed site (sorted by best match) with the following search expressions: “stroke and exosome” = 317 results, 2007-2022; “stroke and extracellular vesicles” = 483 results, 1974-2021; “stroke AND exosome AND intranasal” = 6 results, 2016-2021.
The following numbers specify papers cited in our MS (26-33, 40, 41, 50, 55, 58, 64), referring to studies that used EVs as a neuroprotective strategy in stroke models.
Summarizing, our study presents the following set of methodologies and results:
1) EVs were obtained from human adipose tissue mesenchymal cells through a minimally invasive procedure (liposuction), and obtained from healthy individuals.
2) The model was a focal permanent stroke by thermocoagulation of the pial vessels.
3) The EVs administration was intranasal 24 hours after the stroke.
4) Short- (42 and 72 hours) and long- (7-42 days) term effects caused by stroke and by intranasal EVs administration were evaluated.
5) The increase in the number of EVs in the peri-infarct region compared to other brain regions was observed.
Throughout the text, we discussed the relevance of this set of methodological approaches and results, pointing that we believe that our work presents relevant contributions to the theme.
We added the article cited by the reviewer as a reference in our MS.
Point 2: The timeline and figure 10 are not very clear. Please mention the animals and groups very clearly and explain the timeline from baseline behavior until euthanasia.
Response 2: We thank the reviewer for the observation. Accordingly, aiming to clarify the timeline of the study we modified figure 10 (please, see the attached figure), which was included in the text.
Fig. 10 Experimental timeline. a) Dose curve. EVs intranasal administration was performed 24 hours after surgery (100, 200 or 300 µg/kg). b) Short- and long-term parameters evaluated after EVs (200 µg/kg) intranasal administration 24 h after surgery. Short-term effects: number of EVs in brain regions, BBB permeability and infarct volume; Long-term effects: CT: Cylinder Task; OFT: Open Field Task; NORT: Novel Object Recognition Task; EPMT: Elevated Plus Maze Task; Brain Angiogenesis.
Point 3: Why EVs (especially the ones obtained from patients) were kept separately for treatment and not pooled in together. What if EVs obtained from one patient is more effective than the other patients? Why were EVs obtained only from 2 female patients? At least 3 patients are needed to have minimum randomization and different gender.
Response 3: We thank the reviewer for the comment. Although the collection of human AD-MSC is an easy procedure, during the COVID-19 pandemic the liposuction procedure has been very restricted in Brazil. Because of this situation, we were only able to collect AD-MSC from two female healthy individuals. In our group, we expect to have the opportunity to collect more human AD-MSC in the future.
Thus, we opted to present the results from each individual separately, aiming to show that we obtained EVs presenting characteristics and neuroprotective effects in our stroke model from both sources, when intranasally administered. The suggestion of the reviewer is very noteworthy and the pool of EVs from different individuals will be performed when Brazilian sanitary conditions get better.
We thank the reviewer for the specific and relevant comment concerning the difference between the effectiveness of EVs from the 2 individuals. Most of their effects are similar. The only difference was observed in the OFT (Fig. 6f), in which it was observed that only EV2 was able to recover the OFT long-term memory in the 2nd session OFT. However, in the 3rd session all EVs similarly recovered the OFT long-term memory.
Accordingly, we changed the description of Fig. 6 (page 9 and line 181): "All groups presented normal short-term memory (evaluated only in the first session). Only the ISC group presented impairment of long-term memory (evaluated by comparing the first with the third session), an effect reversed with all EV treatments. However, the recovery of long-term memory evaluated in the second session was observed only with EV2 treatment."
Point 4: Are the authors comparing the therapeutic efficacy of commercial EVs versus EVs from patients? If so, what is the rationale?
Response 4: The commercial human AD-MSC were purchased from the LONZA company (POIETICS Bank Adipose-Derived Stem Cells (cat. #PT-5006, donor 34464, male 58-year-old), which is a consolidated company producing many types of cells used by numerous research groups. Thus we compared the therapeutic efficacy of these renowned commercial EVs with the therapeutic efficacy of EVs administration released by AD-MSC collected by us from healthy individuals. Firstly, all the characterization of the cells and the released EVs were performed simultaneously. All the evaluated parameters were similar among commercial and healthy individual (obtained by us) sources. Then, the next experiments, concerning the evaluation of the neuroprotective effects, were performed with the 3 sources.
Point 5: When the animals were intranasally injected with EVs, was just one nostril used or were they injected into both the nostrils?
Response 5: Thank you for this observation. Accordingly, we clarified this topic by changing the sentence on page 20, line 400, emphasizing that the EVs were administered in both nasal cavities.
“4.3.2 Intranasal EVs treatment: Intranasal EVs treatment was performed 24 h after the ischemic or sham procedure. Sedated animals (O2 flow rate at 0.8-1.0 L/min with Isoflurane levels of 2.5–3.0 %) received a single 50 µL of EVs (ISC+EVs) or 50 µL PBS (Naive, Sham, ISC), slowly administered for 30 secs, in both nasal cavities, as previously reported.”
Overall, this study does not have any novelty described by the authors, the methods and animal grouping are not very clear, and the study is biased by having individual EVs obtained from one gender from only 2 patients. There is no randomization, therefore, this reviewer rejects this manuscript.
R: We hope that all considerations/opinions made above by the authors could modify the decision of the reviewer.

Round 2
Reviewer 2 Report
Thank for answering the reviewer comments, it would have been much appreciated if the changes were made with a different color for easy identification.
Though most of the reviewer comments are convincing, the novelty of the study and the study design is not very appropriate. At the end, the authors have only 2 females and the EVs were used individually in this study. This accounts for bias in the results. This study needs to be expanded with more samples by pooling them. The results as such are not acceptable because of the poor study design.
Author Response
Response to Reviewer 2 Comments:
Thank for answering the reviewer comments, it would have been much appreciated if the changes were made with a different color for easy identification.
Though most of the reviewer comments are convincing, the novelty of the study and the study design is not very appropriate. At the end, the authors have only 2 females and the EVs were used individually in this study. This accounts for bias in the results. This study needs to be expanded with more samples by pooling them. The results as such are not acceptable because of the poor study design.
Dear Reviewer,
Thank you for reading the resubmitted version of our MS.
Below are the answers to the Academic Editor's comments:
The Conclusions should make clear that differences in mouse results between EVs from the two donors and supplied MSCs are not controlled for and that comparison between EV0, EV1 and EV2 are not able to be made due to potential large differences in EVs between donors.
Fig 4(e) and 4(f), should EV3 be EV2?”
We adapted the following paragraphs in the MS:
Page 11, line 286, at the final of the discussion session:
“Here, we demonstrate that intranasal hAT-MSC-derived EVs administered 24 h after stroke promoted long-term neuroprotective effects, offering a remarkably therapeutic window. Although the three EVs show protective properties, findings are not comparable since EV0, EV1, and EV2 are from different sources (2 healthy individuals and commercial hAT-MSCs). Together, these findings indicate that hAT-MSC-derived EVs are a promising potential therapeutic strategy for patients with focal permanent ischemic stroke.”
Page18, lines 592, 4. Material and methods, 4.6. Statistical Analyses
Comparison between EVs (EV0, EV1, and EV2) were not performed due to differences in EVs sources (2 donors and one commercial).
We appreciate the observation and apologize for our mistake in the Fig. 4.
We correctly name Fig. 4(e) and 4(f) for EV2.